# Notch signalling mediates reproductive constraint in the adult worker honeybee

Elizabeth J. Duncan[1],[†], Otto Hyink[1] & Peter K. Dearden[1]

The hallmark of eusociality is the reproductive division of labour, in which one female caste reproduces, while reproduction is constrained in the subordinate caste. In adult worker honeybees (*Apis mellifera*) reproductive constraint is conditional: in the absence of the queen and brood, adult worker honeybees activate their ovaries and lay haploid male eggs. Here, we demonstrate that chemical inhibition of Notch signalling can overcome the repressive effect of queen pheromone and promote ovary activity in adult worker honeybees. We show that Notch signalling acts on the earliest stages of oogenesis and that the removal of the queen corresponds with a loss of Notch protein in the germarium. We conclude that the ancient and pleiotropic Notch signalling pathway has been co-opted into constraining reproduction in worker honeybees and we provide the first molecular mechanism directly linking ovary activity in adult worker bees with the presence of the queen.

[1] Department of Biochemistry, Laboratory for Evolution and Development, Genetics Otago and Gravida (The National Centre for Growth and Development), University of Otago, P.O. Box 56, Dunedin 9054, Aotearoa-New Zealand. † Present address: Department of Biological Sciences, School of Biology, University of Leeds, Leeds LS2 9JT, UK. Correspondence and requests for materials should be addressed to P.K.D. (email: peter.dearden@otago.ac.nz).

Eusocial insects display a remarkable division of labour where one female, the queen, reproduces while the remainder of the females forage and rear brood[1]. In eusocial Hymenoptera, worker reproduction is limited by physiology established during development, although in most species adult workers retain some ability to reproduce[2,3]. The potential for the worker caste to reproduce, particularly in hymenopteran species, generates a source of conflict in social insect societies[4] that has led to the evolution of mechanisms to constrain reproduction in the non-reproductive or 'worker' caste when a queen is present[5,6]. These reproductive constraints are critical to the evolution of eusociality. By negating conflict over worker reproduction[4] and polyandry[7] these mechanisms maintain social harmony[5,6] facilitating the transition from independent reproduction to the situation in eusocial animals, where reproduction is limited to one female caste. Understanding the molecular mechanisms of reproductive constraint is therefore integral to our understanding of the evolution of eusociality.

In honeybees, reproductive constraints are both behavioural and physiological; examples include policing of worker-laid eggs and reduced ovary activity in workers[5]. When the queen is lost or removed from the hive, these constraints are largely eliminated and the worker castes of many Hymenopteran species, including honeybees (Apis mellifera), are able to initiate oogenesis and lay eggs, although there is variation in their ability to do so[2]. In a queen-less environment, if there is no opportunity to make another queen, approximately one-third of honeybee workers activate their ovaries (Supplementary Fig. 1B) and lay eggs[8]. These eggs are unfertilized and haploid, and they generate fertile male offspring.

In adult worker honeybees, reproduction is normally constrained via pheromones produced both by the queen bee[9–11] and her brood[12–14]. These pheromones, including queen mandibular pheromone (QMP), inhibit ovary activation and reproductive behaviour in worker bees[11], as well as inducing young workers to feed and groom the queen, and to perform colony-related tasks[15,16]. How these pheromones inhibit reproduction in the honeybee worker is not well understood, nor is it understood how this constraint is overcome to allow ovary activation and egg-laying in worker bees in the absence of a queen. The phenotypic differences between queen and worker ovaries are established during larval development in response to larvae being fed royal jelly. This nutritional stimulus initiates distinct developmental trajectories in larvae[17], resulting in the morphologically distinct queen and worker castes and the reduced reproductive capacity of workers. Worker bees have a reduced reproductive capacity, they have no spermatheca (sperm storage organ) and their ovaries are smaller[18]. Many studies have focused on how sub-fertility in workers is established during development[17,19] with fewer studies addressing reproductive constraints in adult eusocial insects, which is the focus of this study. A small number of genes have been shown to respond to the absence of the queen in worker bee ovaries, for example see (refs 20–22). These studies have not, however, shown that these changes precede physiological changes in the ovary, nor do they demonstrate that modulation of these pathways can overcome reproductive constraints imposed by exposure to QMP.

In this paper, we demonstrate that an ancient and conserved cell-signalling pathway, Notch cell signalling, acts in the honeybee-worker ovary to represses reproduction. Notch signalling is pivotal during embryogenesis and in adult animals to control processes such as differentiation and cell fate specification and, depending on the biological context, proliferation and apoptosis[23,24]. Notch signalling is typified by its role in specification of neuronal versus epidermal cells during neurogenesis in Drosophila, but Notch signalling has a role in the development of most tissues and organs in many animals[23]. In Drosophila, Notch signalling has multiple roles in oogenesis and reproduction; for instance, Notch signalling is responsible for specifying the germ cell niche[25], controls proliferation and differentiation of somatic follicle cells[26], and defines distinct follicle cell populations[27]. Here, we demonstrate, using a chemical inhibitor of Notch signalling, that inhibition of Notch signalling can overcome the repressive effect of QMP on ovary activity. We also show that the Notch receptor is degraded in the area of the ovary that houses the germ-line stem cells and early oocytes in the absence of the queen. Notch signalling acts on the earliest stages of oogenesis in the germarium, the region of the ovary that has been shown to differ morphologically between queen-right and queen-less worker bees[28]. We conclude that Notch signalling is a proximate mechanism by which QMP represses ovary activity and maintains reproductive sterility in the worker honeybee.

## Results

**Notch cell signalling regulates worker ovary activity**. Notch signalling is a highly conserved cell-signalling pathway with pleiotropic roles in development[29]. In adult Drosophila, Notch signalling has multiple roles in oogenesis[30] including specification and maintenance of the germ-stem cell niche[25,31], a role we hypothesized might be required for oogenesis in adult worker honeybee ovaries. To functionally test this hypothesis we treated newly emerged worker bees with an inhibitor of Notch signalling, DAPT (N-[N-(3,5-Difluorophenacetyl)-L-alanyl]-S-phenylglycine t-butyl ester). Binding of a ligand, such as Delta or Serrate, to the Notch receptor causes it to be cleaved by γ-secretase and a portion (the Notch intracellular domain (NICD)) translocates to the nucleus where it regulates gene expression[29]. DAPT prevents activation of the Notch receptor by inhibiting γ-secretase, and has been used extensively to study the role of Notch signalling in development of a range of animals, including vertebrates (for example, see ref. 32), cnidarians[33] and arthropods[34–38]. Treatment with DAPT causes phenocopies of Notch mutants when fed to Drosophila[39], has been used to examine the role of Notch signalling in segmentation in honeybees[40] and reproduces the phenotype induced by RNA interference against the Notch receptor in the cockroach Periplaneta americana[35] (insects separated by ∼400 million years of evolution[41]).

To determine whether Notch signalling is involved in regulating ovary activity in worker honeybees, we caged newly emerged bees and exposed them to DAPT (or solvent control) for 10 days in the absence of QMP. To minimize the effect of seasonal variation in the propensity of worker bees to activate their ovaries[42] these experiments were performed in mid-summer in the southern hemisphere (December–January). Following treatment, ovaries were dissected, photographed and scored for levels of ovary activity based on a modified Hess scale[43].

Treating bees with DAPT caused a gain-of-function phenotype; there was a significant increase in the proportion of bees that had fully developed eggs in their ovaries (score = 3) and a significant reduction in the proportion of bees that were reproductively inactive (score = 0; Fig. 1a). These data indicate that Notch signalling normally has a role in maintaining bees in a reproductively inactive state.

**QMP represses ovary activity via Notch signalling**. To determine if Notch signalling is the proximate molecular mechanism by which QMP constrains reproduction in adult worker bees, we also treated newly emerged bees with DAPT in the presence of QMP for 10 days. These experiments were carried out in late summer/early autumn (March–April in the southern hemisphere) as this is when the highest proportion of bees activate their

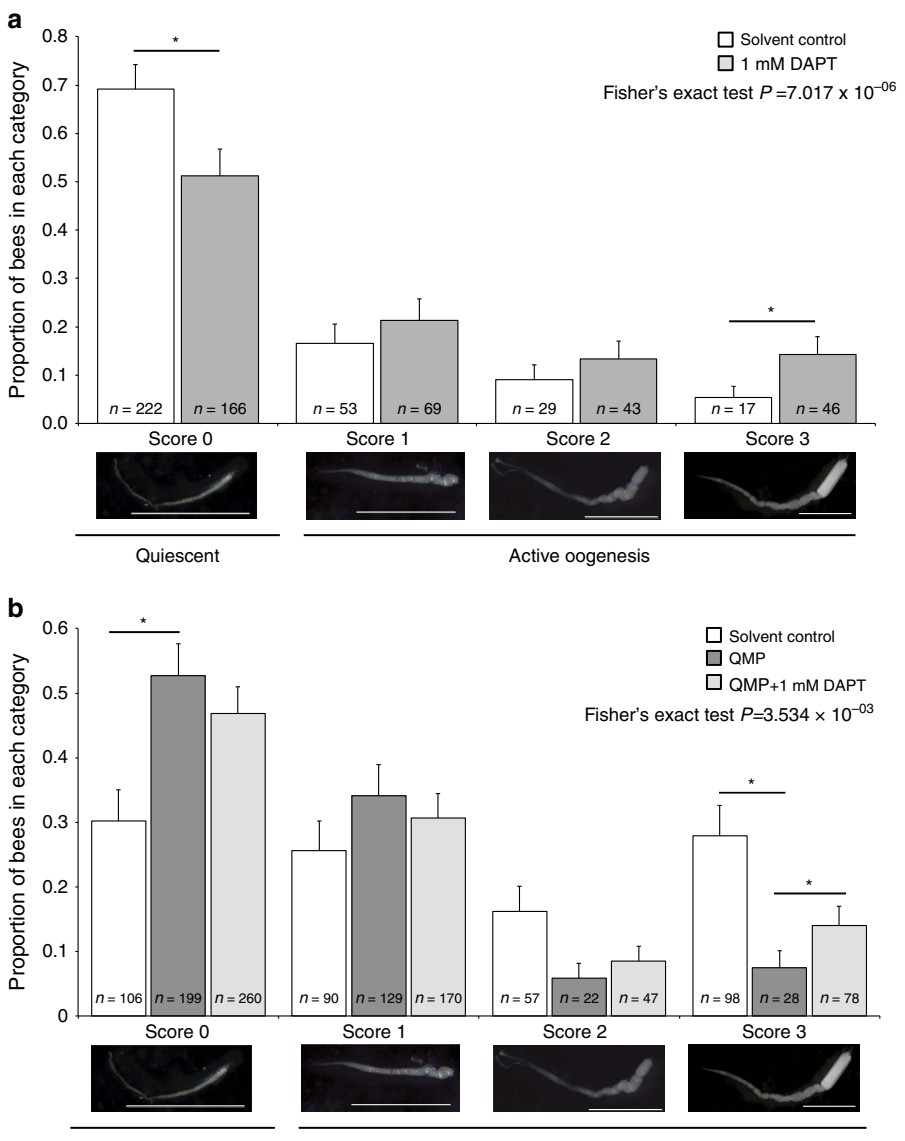

**Figure 1 | Inhibition of Notch signalling promotes ovary activation. (a)** Proportion of bees scored as reproductively inactive (score = 0), and degrees of reproductively active (score = 1–3) following treatment of newly emerged bees for 10 days with 1 mM DAPT ($n=324$) or solvent control ($n=321$). Representative examples of the morphological differences seen between ovary scores are shown beneath the graphs. Treating newly emerged bees for 10 days with DAPT during mid-summer results in a significant decrease in the number of bees that are reproductively inactive (score = 0, 69% reduced to 51%) and a significant increase in the number of bees that are actively laying eggs (score = 3, 5% to 14%). Experiments were performed in triplicate on three separate occasions. **(b)** Proportion of bees scored as reproductively inactive (score = 0), and degrees of reproductively active (score = 1–3) following treatment of newly emerged bees for 10 days with solvent control ($n=351$), QMP and solvent control ($n=378$) or QMP and 1 mM DAPT ($n=555$). In late summer/early autumn there is a higher level of ovary activity in solvent-only treated bees (28% actively laying eggs, compared with 5% in mid-summer, consistent with reported seasonal variation in ovary activity[40]. Exposing these bees to synthetic queen pheromone reduced this ovary activity significantly (compare white bars and dark bars), with the proportion of bees actively laying eggs (score = 3) reduced from 28 to 5%), this inhibition was partially overcome by treatment with DAPT (compare dark bars with light grey bars) and the proportion of bees actively laying eggs increased significantly from 5 to 16%. Experiments with QMP were performed in triplicate on two separate occasions. Error bars are 95% confidence intervals, *non-overlapping 95% confidence intervals.

ovaries[42], allowing us to show that QMP could efficiently repress ovary activity in our experimental system (Fig. 1b, compare white bars with dark grey bars). Indeed, 28% of bees not exposed to QMP were actively laying eggs (ovary score = 3) during this late summer period, and exposure to QMP efficiently repressed this to just 5%.

We could overcome the inhibitory activity of QMP, at least partially, by treating bees with the inhibitor of Notch signalling

DAPT (Fig. 1b, compare dark bars with light grey bars). We observed that the proportion of bees scored as actively laying eggs (score = 3) rose from 5% with just QMP to 15% in the presence of QMP and DAPT. This finding indicates that partial inhibition of Notch signalling is able to overcome the reproductive constraints conferred by QMP, and demonstrates that Notch signalling may be a key mechanism by which QMP constrains reproduction in adult worker bees.

**Loss of the queen reduces Notch signalling in worker ovaries.** Having shown that blocking Notch signalling activates reproduction in caged worker bees, we then wanted to determine if changes in Notch signalling could be identified in the honeybee ovary which might indicate a direct role for Notch signalling in controlling worker bee reproduction.

Signalling through the Notch pathway causes transcriptional activation of a range of target genes, including genes of the E(spl)-C (enhancer of split complex)[44,45]. We measured the expression of the four genes of the honeybee E(spl)-C, using quantitative RT–PCR (qRT–PCR), as a proxy for the activity of Notch signalling in the ovaries of queen-right workers (workers in a hive containing a queen), queen-less workers and queen bees (Fig. 2). As previously mentioned, ovary development in adult queen-less worker ovaries transitions from quiescent ovaries (score = 0) to fully active egg-producing ovaries (score = 3), and we measured

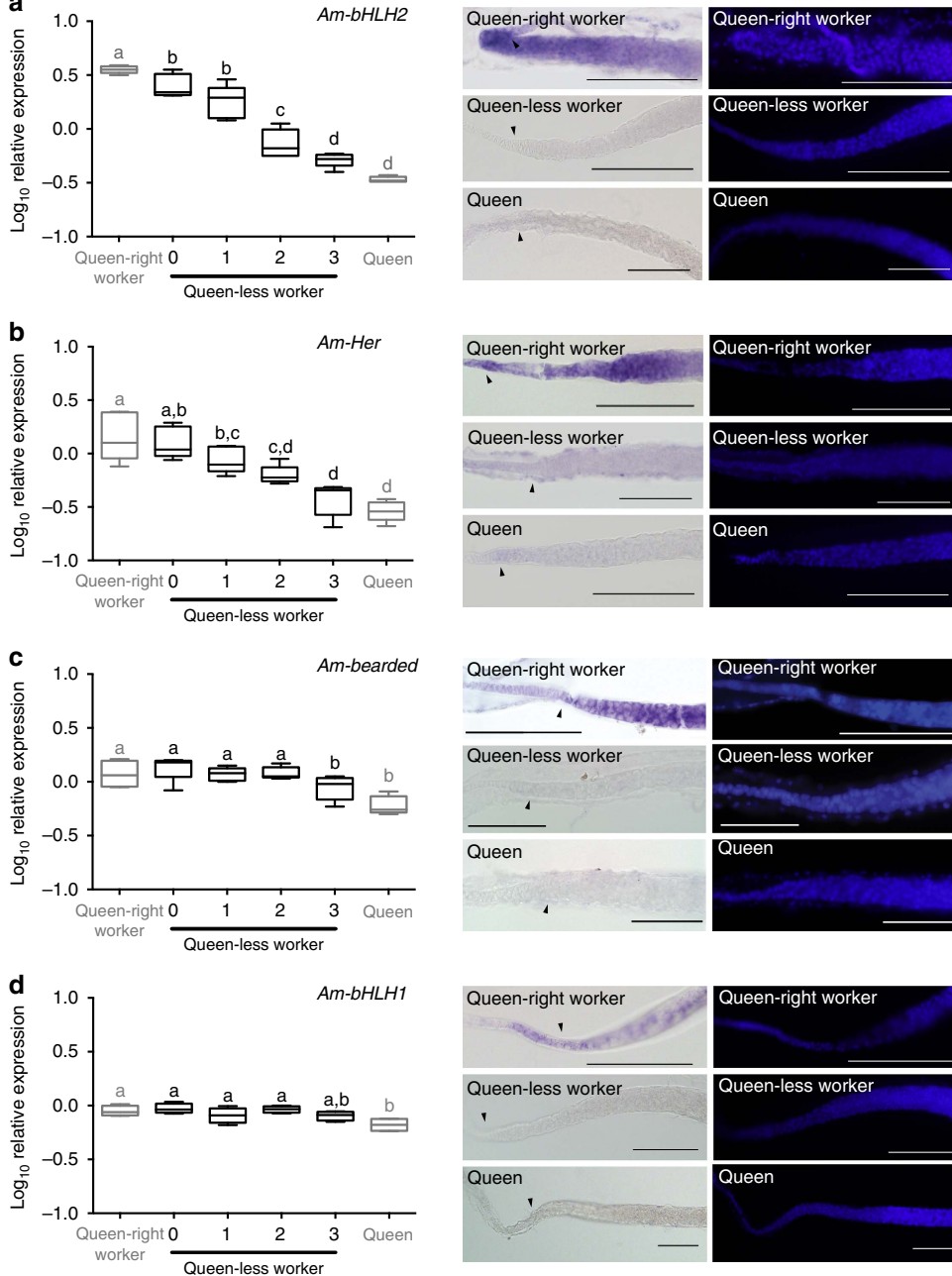

**Figure 2 | Notch-responsive genes are altered in the ovaries of queen-less bees.** Expression of genes of the four genes of the E(spl)-C were examined using qRT–PCR and *in situ* hybridization. Expression of two of these genes, *bHLH2* (**a**) and *Her* (**b**), decrease significantly as the bees respond to the loss of the queen and activate their ovaries, while the expression of the other two genes of the complex, *bearded* (**c**) and *bHLH1* (**d**) do not vary significantly. qRT–PCR data is the mean of transcript levels (Log$_{10}$) in five biological samples for each condition. Boxplot whiskers indicate minimum and maximum, the box is defined by 25th percentile, median and 75th percentile. Differences in target gene expression were determined by analysis of variance with a Tukey's *post hoc* test, statistical differences ($P < 0.05$) are denoted by different letters. Using *in situ* hybridization to determine which cells of the ovary are responding to a Notch signal shows that cells of the germarium of queen-right worker bees express all four genes of the E(spl-C) (**a–d**), whereas the same region of the ovary in queen-less worker and queen bees do not express mRNA for these genes. Following *in situ* hybridization ovaries were counter-stained with 4',6-diamidino-2-phenylindole (DAPI; right panels). Scale bars, 100 μm. Arrow heads indicate the border between cells of the terminal filament and germarium.

expression of the E(spl)-C genes across these stages of ovary activity (Fig. 2). Expression of two of the E(spl)-C genes, *bHLH2* and *Her,* decreases as the ovaries become progressively more active, implying that Notch signalling activity decreases as the bees respond to the absence of QMP and initiate oogenesis (Fig. 2). Studies of the *Drosophila* E(spl)-C indicate that genes within the complex are regulated by discrete enhancers and can be differentially responsive to Notch signalling[46–48]. Similar differences in regulation may also account for the differential expression of *bHLH2* and *Her*, but not *bearded* or *bHLH1* in the honeybee ovary (Fig. 2). Decision tree recursive partitioning demonstrated the expression of *bHLH2* alone correctly, andclassified the physiological state of the ovaries (as either active or inactive) 97% of the time (Supplementary Table 1), indicating that the expression of this gene is closely related to ovary activity in worker bees. *bHLH2* expression also decreases before there is any detectable difference in morphology (compare queen-right worker ovaries with queen-less worker ovaries (score = 0, Fig. 2a)). That *bHLH2* expression changes before any detectable difference in morphology raises the possibility that Notch signalling is downregulated early in ovary activation, and may directly link QMP exposure with ovary activity in the adult worker honeybee.

Insect ovaries are made up of multiple cell types. At the anterior of each ovariole lies the cells of the terminal filament, these are followed by the germarium which houses the germ-line stem cells, somatic stem cells, gonia and early cysts (Supplementary Fig. 1). As our qRT–PCR experiments (Fig. 2) were performed on RNA derived from whole ovaries and are thus an amalgamation of RNA expression in all ovarian cell types, it was important to identify the region/s in which Notch signalling was acting in the ovaries to constrain reproduction (full expression patterns are shown in Supplementary Fig. 2). To determine which cells in the honeybee ovary were receiving the Notch signal we used the expression of the E(spl)-C genes as a proxy for Notch activity, using *in situ* hybridization to visualize cells expressing RNAs from these genes (Fig. 2, Supplementary Fig. 2). The E(spl)-C is a complex of Notch-responsive genes that is evolutionarily conserved among insects and crustaceans[45,49]. In most insects and crustaceans, the complex consists of three basic helix–loop–helix transcription factors and a bearded class protein[45,49]. In honeybees, it is unclear what the functions of these four genes are[45].

*In situ* hybridization reveals expression of all four genes of the E(spl)-C in queen-right worker ovaries within the cells of the germarium, the region of the worker ovary previously linked to differences in fertility[28] and where oocytes are specified from presumptive germ-stem cells. All four genes of the E(spl)-C are also expressed in the posterior of the ovariole as the oocyte matures, with RNA detected in the nurse and follicle cells (Supplementary Fig. 2). The expression of these genes in regions of the ovary other than the germarium may explain why, using qRT–PCR on whole ovaries, the expression of *bHLH2* and *Her* decreases markedly as worker ovaries initiate oogenesis while the other two genes of the complex do not respond (Fig. 2).

The expression of the E(spl)-C genes in the germarium indicates that these cells are receiving a Notch signal. In contrast, we see no expression of these genes in the equivalent regions of queen-less or queen ovaries (Fig. 2), indicating that the cells within the germaria of bees undergoing active oogenesis (queen-less worker and queen bees) are not receiving a Notch signal.

The differences in E(spl)-C gene expression in the germaria links Notch signalling with differential fertility in the honeybee. The expression of the E(spl)-C genes is consistent with our inhibition of Notch signalling (Fig. 1); worker bees in queen-right hives have active Notch signalling in the germaria of their ovaries, while this signalling is absent from queen-less workers and queens. Downregulation of Notch signalling in the germarium is thus associated with an active ovary.

**Mechanisms for the reduction in Notch signalling activity**. To determine how Notch signalling is regulated differently between the ovaries of queen-less and queen-right worker bees, we examined the expression of the Notch ligands, *Delta* and *Serrate*. Both *Delta* (Fig. 3a) and *Serrate* (Fig. 3b) transcripts are differentially regulated in queen-less worker ovaries, but only *Delta* RNA is detected in the germarium. In the queen ovary, *Delta* RNA is expressed by all cells in the germarium but appears enriched in a specific cell type, probably the dividing germ cells (cystocyte clusters), consistent with previously published data[50]. *Delta* RNA is, however, expressed uniformly in all cells of the germarium of queen-right worker and queen-less worker ovarioles (Fig. 3a), implying that *Delta* expression does not account for the differences in Notch activity observed between queen-right and queen-less worker ovaries (Fig. 2).

Since expression of Notch ligands does not explain differential activation of Notch signalling in the honeybee ovary, we examined the expression and localization of the Notch receptor. We used an antibody to the intracellular domain (NICD) of *Drosophila* Notch, which cross-reacts with honeybee Notch[50] (Supplementary Fig. 3), to visualize its subcellular localization in the germaria of queen-right, queen-less and queen ovarioles (Fig. 4a). In cells that are not actively receiving a Notch signal, we expect immunoreactivity for the NICD to co-localize with filamentous actin underlying cell membranes, consistent with the transmembrane location of the Notch receptor. Activation of the Notch protein causes the NICD to translocate to the nucleus where we would expect to see co-localization with nuclear markers.

Differences in the subcellular distribution of the Notch receptor were observed between germaria from ovarioles of queen-right workers, and those from queen-less workers and queen bees. In queen-right worker ovaries (Fig. 4a), the NICD co-localized with nuclei in the terminal filament and nuclei of cells of the germarium, indicating that these cells have active Notch signalling, consistent with the expression of the E(spl)-C genes in the germarium of queen-right worker bees (Fig. 2). In queen-less worker and queen ovaries, the Notch receptor is present only in the nuclei of terminal filament cells (Fig. 4a), with little NICD immunoreactivity throughout the anterior of the germarium, consistent with the lack of E(Spl)-C expression in these cells (Fig. 2). The lack of immunoreactivity in the anterior germarium indicates that these cells are unable to respond to the Notch ligands due to the absence or low levels of the Notch receptor on their membranes. Notch protein is also not detected in the cytoplasm of these cells where it would indicate recycling or intracellular processing of the Notch protein (as seen in follicle cells late in oogenesis, Supplementary Fig. 4, where immunoreactivity for the Notch protein and expression of genes of the E(spl)-C may indicate that Notch signalling has a role in supporting active oogenesis, specifically by patterning the follicle cells (refer to Supplementary Note 1)). In queen-less workers, the Notch receptor is detected on the membranes of cells further down the germarium, where the actin-rich polyfusome structures give rise to ring canals, indicating that these cells are able to, but are not, receiving a Notch signal.

The absence of the Notch receptor in the cells of the germarium of queen-less worker bees provides a mechanism for the differences in Notch activity we observed between queen-less and queen-right worker bees. In the absence of the queen, degradation of the Notch receptor in these cells would render

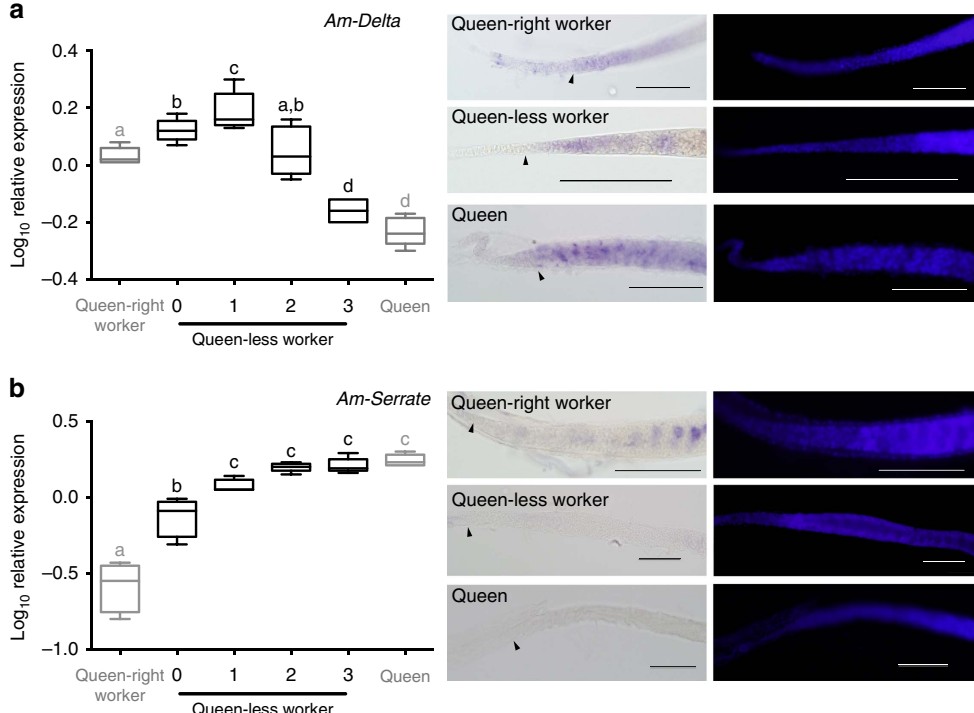

**Figure 3 | Differences in Notch activity are not associated with expression of ligands.** (**a**) *Delta* transcript levels increase transiently in queen-less worker bees as the ovaries begin to undergo cell differentiation, but transcript levels decline as vitellogenesis (deposition of yolk) begins. *In situ* hybridization reveals that *Delta* RNA is expressed throughout the germarium of queen-less, queen-right and queen ovarioles. (**b**) *Serrate* RNA levels increase steadily in the ovaries of queen-less worker bees, but *in situ* hybridization reveals that *Serrate* is not expressed in the germarium of queen-less, queen-right or queen ovarioles. qRT–PCR data are the mean of transcript levels (Log$_{10}$) in five biological samples for each condition. Boxplot whiskers indicate minimum and maximum, the box is defined by 25th percentile, median and 75th percentile. Differences in target gene expression were determined by analysis of variance with a Tukey's *post hoc* test, statistical differences ($P < 0.05$) are denoted by different letters. Following *in situ* hybridization ovaries were counter-stained with DAPI (right panels). Scale bars, 100 μm. Arrow heads indicate the border between cells of the terminal filament and germarium.

these cells refractory to Notch signalling. The antibody we used to detect the Notch receptor recognizes the intracellular domain of Notch[51], which can be depleted through interactions with Numb[52]. *Numb* RNA is induced twofold in the ovaries of queen-less worker bees (Fig. 4b). Induction of *numb* occurs early in the ovary-activation process (score = 0), when queen-less and queen-right worker ovaries are morphologically indistinguishable. This early and sustained upregulation of *numb* RNA in response to the absence of the queen raises the possibility that Numb may target the Notch receptor for degradation in the germaria of queen-less worker bees. Consistent with this hypothesis, we detected *numb* RNA in cells of the germaria of queen-less worker bees and queen bees, whereas *numb* RNA is barely detectable in the germaria of queen-right worker bees (Fig. 4c). In queen-less worker bees *numb* RNA is detected in all the cells of the germarium, including the dividing germ cells (cystocyte clusters). The upregulation of *numb* RNA coincides spatially and temporally with loss of immunoreactivity for the NICD, and the loss of expression of Notch target genes, with no effect on the levels of Notch mRNA expression (Supplementary Fig. 5). These data, consistent with our functional studies, imply that, in the absence of the queen, Numb may degrade the Notch receptor, relieving the direct repressive effect of Notch cell signalling in the germarium, allowing activation of the ovary and thus worker reproduction (Fig. 4d).

## Discussion

The existence of sterile castes in eusocial insects confounded Charles Darwin, who called it the 'one special difficulty' in his theory of evolution by natural selection[53]. Eusociality requires one female caste to have evolved reproductive dominance and reproduction in the other female caste to be constrained. The processes that constrain reproduction can be behavioural or physiological, and it has been argued that they have evolved as the result of an evolutionary 'arms race' between queen and worker castes over worker reproduction[54].

Our data indicate that, in honeybees, QMP inhibits reproduction by stimulating Notch signalling in worker bee ovaries in the region where germ cells are specified (Fig. 4d). It has yet to be determined whether QMP is directly affecting the ovary or acting via signalling between the brain or antennae (indicated by the dashed line linking QMP with *numb* in Fig. 4d). Repression of oogenesis via Notch signalling in the germaria of worker bee ovaries is consistent with the low levels of Notch activity we observe in the queen ovaries (Figs 2–4) where oogenesis is actively occurring[18]. The association of active Notch signalling with repression of oogenesis contrasts with roles for Notch signalling in controlling reproduction in *Drosophila*[30]. In *Drosophila*, Notch signalling both establishes and maintains the germ-stem cell niche[25,31]. Increased Notch signalling results in a larger niche and more stem cells[25,55], but it is unclear whether this role is ancestral and representative of all insects. Notch signalling has other roles in *Drosophila* oogenesis including promoting the differentiation of follicle cells[30] but in other insects[56,57], Notch signalling inhibits follicle cell differentiation, implying that the ancestral function of Notch signalling in insect follicle cells is to maintain undifferentiated cell fates[56,57]. Our data may indicate that the ancestral function of Notch signalling in the germarium was also to repress cell differentiation.

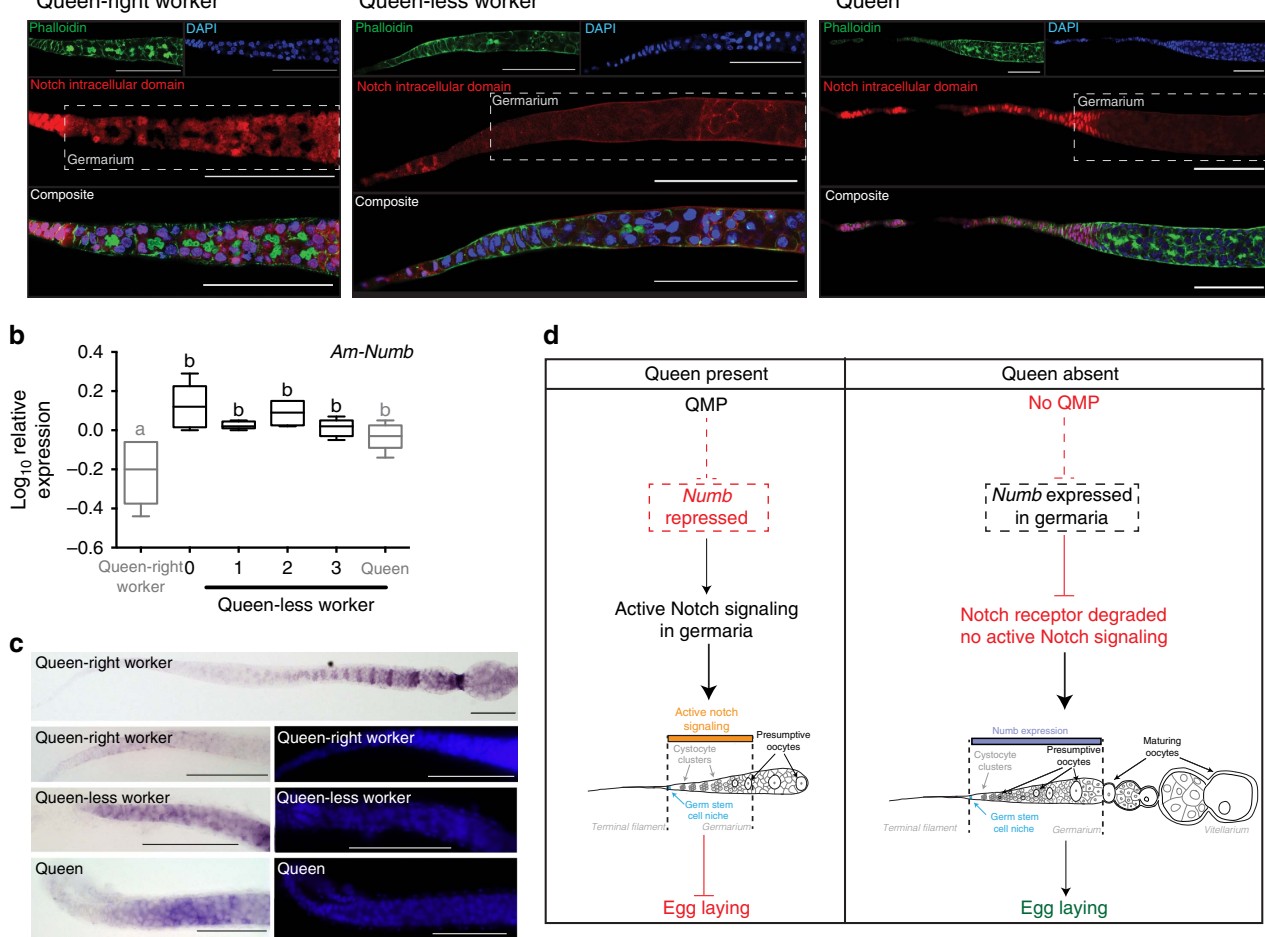

**Figure 4 | The Notch receptor is differentially localized in worker ovaries.** Activation of the Notch receptor causes part of the receptor to become cleaved (NICD) where it moves to the nucleus to regulate gene expression. (**a**) Immunohistochemistry using an antibody against the NICD indicates that in queen-right worker bees the NICD is predominately nuclear-localized in the cells of the terminal filament and all of the cells of the germarium. The NICD is also present in the nucleus of the anterior terminal filament cells in queen-less worker bees; but the NICD is essentially absent from the anterior germarium. In the posterior germarium, as oocytes become clearly identifiable, the NICD is detectable on cell membranes but not in the nuclei. In queen ovarioles, the NICD is present in the nucleus of cells of the terminal filament, indicating that these cells are receiving a Notch signal, but there is little immunoreactivity detected throughout the germarium. Ovaries were counter-stained with DAPI and phalloidin to visualize nuclei and cortical actin. (**b**) We examined the expression of *Numb*, a gene implicated in regulation and recycling of the Notch receptor, using qRT–PCR and *in situ* hybridization. qRT–PCR indicates that *numb* mRNA is induced more than twofold when the queen is removed from the hive before any morphological difference is detectable in the ovaries. qRT–PCR data is the mean of transcript levels ($Log_{10}$) in five biological samples for each condition. Boxplot whiskers indicate minimum and maximum, the box is defined by 25th percentile, median and 75th percentile. Differences in target gene expression were determined by analysis of variance with a Tukey's LSD *post hoc* test, statistical differences ($P < 0.05$) are denoted by different letters. (**c**) *Numb* mRNA is barely detectable in the germarium of queen-right worker bees, but is readily detectable in the germarium of queen-less and queen ovarioles. Following *in situ* hybridization ovaries were counter-stained with DAPI (right panels). Scale bars, 100 μm. Arrow heads indicate the border between cells of the terminal filament and germarium. (**d**) Proposed model summarizing the role of QMP in regulating oogenesis via Notch signalling in the honeybee-worker ovary.

Our results show that Notch signalling has a pivotal role in repressing reproduction in the worker honeybee in the presence of the queen and her pheromone. By enforcing the reproductive division of labour and mitigating conflict over male production, Notch signalling is critical to the reproductive constraints that underpin the evolution of eusociality. Notch signalling, presumably as a result of adaptive evolution, has been co-opted from a role in solitary insects into constraining reproduction in the worker honeybee. Notch signalling's fundamental role in the ovary, known to be environmentally responsive[58], has been modified and transformed in honeybees into social control of reproduction in the adult honeybee.

Our data demonstrate that the evolution of reproductive constraint in honeybees is not simple degeneration of worker reproductive potential[5], but targeted control of worker fertility through co-option of a conserved cell-signalling pathway.

## Methods

**Honeybee culture and tissue collection.** *Apis mellifera* were cultured using standard techniques in Dunedin, New Zealand[59]. Honeybees were housed in wooden Langstroth hives or wooden nucleus boxes at three sites around Dunedin. All bees were originally obtained from Betta Bees Research Limited. Queen tissue was obtained from young queens ($\sim$12 months old) that were active egg layers and were heading colonies. All Queen-right worker bees were obtained from standard hive with a laying queen (queen-right). To obtain queen-less worker bees frames with brood and bees were removed from queen-right hives and placed into a standard nucleus box. Colonies were regularly surveyed for queen cells (which were destroyed) and the presence of worker-laid eggs. Ovaries were dissected from queen-less worker bees 2–4 weeks after establishing a queen-less hive; a sufficient period for brood to have emerged, and worker-laid eggs were detected. Stages of

oogenesis were based on an established scheme[50]. For RNA extraction, ovaries were dissected into phosphate-buffered saline (PBS) and examined under a Leica M50 stereomicroscope to determine the levels of ovary activity before snap-freezing on dry ice and storing at −80 °C. The number of individuals dissected for each biological replicate were as follows: queen ($n = 1$), queen-less worker ovary score = 3 ($n = 5$), queen-less worker ovary score = 2 ($n = 10$), queen-less worker ovary score = 1 ($n = 20$), queen-less worker ovary score = 0 ($n ≥ 20$) and queen-right worker ovaries ($n = 40$).

**Classification of ovary activity in queen-less worker bees.** The levels of ovary activity in queen-less worker bees was determined based on a modified Hess scale[43]. Ovaries that were thin, lacked defined ova and were morphologically indistinguishable from queen-right worker ovaries were scored as 0, ovaries that were slightly thickened, showing signs of differentiated cells but with no deposition of yolk were scored as 1, ovaries with clearly defined oocytes and yolk deposition were scored as 2 and ovaries with at least one oval fully mature ova were scored as 3 (ref. 60; Supplementary Fig. 1).

***In vivo* inhibition of Notch signalling.** Frames containing emerging brood were removed from multiple hives and incubated at 35 °C overnight, newly emerged bees were randomly assigned to wooden cages, and cages were randomly assigned to a treatment group. Cohorts of newly emerged bees were raised in $8 × 8 × 4$ cm wooden cages ($n = 100–120$ bees per cage) at 35 °C, with a piece of empty comb attached to the rear wall of the cage. Bees were fed high-protein pollen cake and water was given *ad libitum*. Synthetic QMP was obtained from PheroTech (Delta, BC, Canada) in the form of commercially available strips (BeeBoost). One strip in an average colony is reported to provide effective queen replacement for up to 3 weeks, therefore it was deemed unnecessary to replace the QMP strips during the experiment. Dead bees were removed from cages as soon as they were discovered. DAPT ($N$-[$N$-(3,5-Difluorophenacetyl)-L-alanyl]-S-phenylglycine t-butyl ester) is an inhibitor of γ-secretase[61], treatment with DAPT phenocopies Notch mutants in a range of vertebrate and invertebrate species[32,35,39]. DAPT (Sigma-Aldrich) was dissolved in ethanol and mixed into the food at a final concentration of 1 mM, a concentration known to be effective in phenocopying Notch mutations in *Drosophila* larvae[39] and an equivalent amount of ethanol was added to the control diet. Food was made fresh and bees were fed daily, food intake and lethality were recorded (Supplementary Fig. 6). In experiments without QMP, eggs were routinely observed in the comb after several days, but policing behaviour was also occasionally seen and hatched larvae were never observed in either the DAPT or control treatments. After 10 days, bees were killed, ovaries were dissected and photographed using a Leica Mz75 stereomicroscope with a DFC280 digital camera and Leica Application Suite software (v. 2.5.0.R1). Photographs were randomized and scored blindly by two people, using the scale described above (Supplementary Fig. 1B). Experimental treatments were carried out in triplicate on at least two independent occasions. Differences between control and treated cages were determined using a Fisher's exact test for proportions of each ovary-activation class between treatments. Confidence intervals for proportions were calculated using standard methods.

**Quantitative RT–PCR.** RNA was extracted from snap-frozen ovaries using Trizol reagent (Invitrogen), followed by purification and on-column DNAse treatment using RNAeasy columns (Qiagen). 1 µg of RNA was reverse-transcribed using VILO reagent (Invitrogen) and a 1:10 dilution of this cDNA was used as a template for qRT–PCR. Oligonucleotide primers were designed using Primer3plus[62] and evaluated using Beacon Designer (PREMIER Biosoft). Where possible, primers were designed to span intron/exon boundaries to detect amplification from contaminating genomic DNA. *In silico* specificity of the PCR primers were assessed with primer-BLAST[63]. Oligonucleotide primer sequences used in this study are provided in Supplementary Table 2. PCR products were directly sequenced to confirm the specificity of the amplification reactions. qRT–PCR was carried out on a BioRad CFX Real-Time PCR detection system with SsoFast EvaGreen PCR mastermix, 5 ng of cDNA and 300 nM of each primer. For each condition (queen-right worker, queen-less worker score = 0, score = 1, score = 2, score = 3 and queen ovaries) gene expression was measured for five biological replicates and each measurement was made in duplicate. Expression of target genes was normalized by the geometric mean of the relative quantities for two reference genes that we had determined were stably expressed among our samples: *Rpn2* and *mRPL44* (Supplementary Fig. 7 and Supplementary Note 1). Data was $Log_{10}$ transformed and differences in target gene expression were determined by analysis of variance with a Tukey's *post hoc* test.

**Reference gene identification.** Nine putative reference genes were compared to determine the appropriate reference genes to normalize gene expression measured by qRT–PCR. Reverse transcription and qPCR were carried out as detailed above. Briefly, raw Cq values were obtained for 18 ovary samples ($n = 3$ for queen, queen-less worker scored 0, 1, 2 and 3, and queen-right worker) and used to determine gene expression stability with geNormPLUS. Gene expression stability analysis was carried out with the geNorm algorithm[64] implemented in qbase+ (version 2.6) (ref. 65). geNorm calculates the average pairwise variation of a candidate reference

gene with all other control genes, reported as the *M*-value. The lower the *M*-value, the more stably expressed the gene. The use of a single reference gene for data normalization is not recommended[64], and geNorm also performs a pairwise variation analysis (*V*-value), based on the geometric mean of all the candidate reference genes, to identify the optimal number of reference genes required (Supplementary Fig. 7).

**Decision tree recursive partitioning.** To identify which of the genes analysed in this study was most associated with the reproductive state of the ovary we used decision tree recursive partitioning implemented in Weka[66]. This approach has previously been used to classify tumour samples[67,68] as well as to examine the association of gene expression with ovary activity in the honeybee[20].

***In situ* hybridization and immunohistochemistry.** *In situ* hybridization was carried out as previously described[59,69] briefly, ovaries were dissected from adult bees and fixed in a 1:1 mix of 4% formaldehyde:heptane in PBS. Tissue was rinsed three times in ice-cold methanol before storing at −20 °C. For *in situ* hybridization ovaries were rehydrated through a methanol:PTw (PBS + 0.1% Tween 20) series before dissecting individual ovarioles from the ovaries using fine forceps. Ovarioles were treated with 20 µg of proteinase K before washing in PTw and refixing in 4% formaldehyde. Samples were rinsed before adding hybridization buffer (50% deionised formamide, $4 ×$ SSC, $1 ×$ Denhardt's solution, 250 µg ml$^{-1}$ yeast tRNA, 250 µg ml$^{-1}$ boiled salmon sperm DNA, 50 µg ml$^{-1}$ heparin, 0.1% Tween 20 and 5% dextran sulfate). Samples were prehybridised at 52 °C for a minimum of 2 h. Sense and antisense digoxigenin (DIG)-labelled riboprobes were generated by run-off transcription from cloned and sequence-verified plasmids using the appropriate RNA polymerase and DIG labelling mix (Roche Applied Science). Plasmid clones for *Am-Delta*, *Am-bHLH1*, *Am-bHLH2*, *Am-bearded* and *Am-Her* have been previously published[40,45]. Orthologues of other genes were identified by BLAST analysis[70] (refer to Supplementary Table 2 for primer sequences). Riboprobes were digested with an equal volume of carbonate buffer (120 mM $Na_2CO_3$, 80 mM $NaHCO_3$, pH 10.2). Samples were hybridized at 52 °C overnight followed by washing 6 times at 52 °C with wash buffer (50% formamide, $2 ×$ SSC, 0.1% Tween 20), the final wash was left on overnight. Samples were then rinsed in PTw and blocked with PTw containing 0.1% bovine serum albumin. This was replaced with a 1 in 1,000 dilution of the anti-DIG alkaline-phosphatase-conjugated antibody (Roche Applied Science) and incubated at room temperature for 90 min. Excess antibody was removed by washing with PTw. Samples were transferred to alkaline phosphatase staining buffer (100 mM Tris 9.5, 100 mM NaCl, 5 mM $MgCl_2$ and 0.1% Tween 20) for 5 min before adding fresh buffer containing the reagents nitro blue tetrazolium chloride and 5-bromo-4-chloro-3-indolyl-phosphate (Roche Applied Science). Colour was allowed to develop in the dark, but was periodically monitored under a light microscope. Following staining, ovaries were counter-stained with DAPI and visualized using an Olympus BX61 microscope with a DP71 digital camera. Controls samples hybridized with sense riboprobes are shown in Supplementary Fig. 8. *In situ* hybridization was carried out on at least three independent occasions with ovary tissue obtained from at least five individuals on each occasion.

To validate the C17.9C6 Notch antibody, which is raised against the *Drosophila* NICD, for use in the honeybee we carried out western blotting (Supplementary Fig. 3). As a positive control, *Drosophila* embryos of 4–6 h of age were collected from agar plates and lysed in TNE buffer (10 mM Tris pH 7.4, 100 mM NaCl, 1 mM EDTA) with cOmplete Protease Inhibitors (Roche Applied Science). Ovaries were dissected from mature *Apis mellifera* workers ($n = 25$) and lysed in TNE with protease inhibitors. Protein concentrations were estimated using the Qubit fluorometer (Invitrogen) and Qubit protein assay kit (Invitrogen). Ten micrograms of protein was separated on a 4–12% Novex NuPAGE gel (Invitrogen) at 175 v for 30 min in MOPS buffer (50 mM MOPS, 50 mM Tris Base, 0.1% SDS, 1 mM EDTA and pH 7.7). Proteins were transferred to nitrocellulose membrane in Towbin buffer (25 mM Tris, 192 mM glycine, 0.1% SDS, pH 8.3, 20% methanol), the membrane was blocked in 2% bovine serum albumin in TBS-T before incubation with the C17.9C6 antibody (1:1,000) at 4 °C overnight. Secondary antibodies (1:1,000) and chemiluminescent detection was carried out as per standard protocols. After detection of the Notch signal the membrane was stripped before incubation with the anti-tubulin antibody (E7, 1:1,000).

For immunohistochemistry, ovaries were dissected into PBS. Individual ovarioles were dissected and the membrane covering each ovariole removed from queen and queen-less worker ovarioles. For queen-right workers ovarioles were separated as much as possible. Dissected ovarioles were fixed for 10 min in a 1:1 mix of 4% formaldehyde:heptane in PBS, rinsed three times in 0.1% PTx (PBS + 0.1% Triton X-100) and left to permeabilize for 2 h. Ovarioles were blocked for 30 min in PBTX (PBS + 0.1% Triton X-100 and 0.05% bovine serum albumin). The Notch antibody was obtained from the Developmental Studies Hybridoma Bank and were used at 1:50 dilution, the secondary antibody (goat anti mouse Alexa Fluor 637 (Invitrogen) was used at a 1:200 dilution. Nuclei were counter-stained with DAPI, and cortical actin was stained with Alexa Fluor 488 conjugated phalloidin (Invitrogen). Controls consisted of ovarioles incubated with only the secondary antibody and ovarioles incubated with other primary antibodies. All controls were performed and visualized at the same time as experimental samples. Ovarioles were mounted in ProLong Gold (Invitrogen) and observed on an

Olympus FluoView 1000 confocal microscope. Immunohistochemistry was carried out on at least five independent occasions with ovary tissue obtained from at least five individuals on each occasion.

**Statistical analysis.** Statistical analysis was undertaken using Prism (GraphPad Software Inc). To detect differences in levels of ovary activity following exposure to QMP and DAPT a Fisher's exact test was used. Data are presented as average of the proportions of bees with different levels of ovary activity (0, 1, 2 and 3; refer to Supplementary Fig. 1) across the experiments; error bars are 95% confidence intervals.

Analysis of variance with a Tukey's *post hoc* test was used to determine if there were differences in gene expression between groups. Data were Log$_{10}$ transformed and normality of the data was assessed using a Shapiro–Wilk's normality test. The Brown–Forsythe test was used to determine if there were differences in the variance between groups of data.

Kaplan–Meier survival curves were calculated in Prism (GraphPad Software Inc). To determine if there was any effect of treatment on survival Kaplan–Meier survival curves (Supplementary Fig. 6) were compared using a log-rank test.

**Data availability.** The authors declare that all data supporting the findings of this study are available within the article and Supplementary Information files or are available from the corresponding author on request.

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

## Acknowledgements

This work was funded by a Gravida: National Centre for Growth and Development grant to P.K.D. (MP04). E.J.D. was funded by a University of Otago Division of Health Sciences Career Development Postdoctoral Fellowship. The Notch antibody (C17.9C6) developed by S. Artavanis-Tsakonas was obtained from the Developmental Studies Hybridoma Bank developed under the auspices of the NICHD and maintained by The University of Iowa, Department of Biology, Iowa City, IA 52242, USA. The anti-Rab11 antibody was a gift from Dr Akira Nakamura (RIKEN Center for Developmental Biology). We thank Abigail Romeril for providing PCR primers for bHLH1 and Gemma Palmer for her initial work establishing techniques for raising bees in the laboratory. We also acknowledge Mackenzie Lovegrove, Tamsin Jones and Tahlia Whiting for general laboratory assistance and Professor Martin Beye for critical comment on this manuscript.

## Author contributions

E.J.D. and P.K.D. conceived and designed the experiments. All the authors performed the experiments. E.J.D. and P.K.D. analysed the data, and wrote and edited the manuscript. All authors reviewed the manuscript before submission.

## Additional information

**Competing financial interests:** The authors declare no competing financial interests.

**How to cite this article**: Duncan, E. J. *et al.* Notch signalling mediates reproductive constraint in the adult worker honeybee. *Nat. Commun.* 7:12427 doi: 10.1038/ncomms12427 (2016).

