## [Peer Review File · Nature Communications]

Reviewers' comments:

Reviewer #1 (Remarks to the Author):

As one of the eusocial insects, the honeybee workers retain their ovaries but their development is inhibited by the presence of the queen. The absence of the queen and brood activates the ovaries of adult worker bees to lay haploid male eggs, but the underlying molecular mechanisms remain unclear. This study shows that Notch signaling is the missing link between the ovary activity in adult worker bees and the presence of the queen. The authors have provided several pieces of experimental evidence: 1) Inhibiting Notch signaling by DAPT activates the development of the worker's ovaries even in the presence of the queen; 2) Notch signaling is active in the germ cells and somatic cells of the worker's ovaries in the absence of the queen based on the expression of Notch intracellular domain and Notch response genes; 3) numb mRNAs are highly expressed in the ovaries of the queen, but not the in the ovaries of the workers in the presence of the queen; 4) Delta, encoding a Notch ligand, is expressed in the ovaries of both the queen and the worker. Overall, the findings are interesting and exciting, and should be appropriate for publication in Nat Comm.

Before considering its publication, the authors need to address the following comments to further strengthen this paper. First, it is important to know which cells express the Notch ligand Delta using fluorescent mRNA in situ and DNA staining. Germ cells have bigger nuclei than somatic cells. Second, based on the results in Figure 4, numb mRNA should be expressed in germ cells. The authors should use fluorescent mRNA in situ and DNA staining to verify it. Third, the authors should clarify if the eggs laid by the DAPT-treated workers can develop normally.

Reviewer #2 (Remarks to the Author):

The authors provide convincing evidence that an evolutionarily conserved intracellular signaling pathway has been deployed to control the fertility status of adult honey bee queens and workers. Specifically, they show that the Notch signaling pathway is active in the upper germarium of workers kept in the presence of a queen or its synthetic pheromone equivalent (QMP), while it is inactivated in queenless workers or egg-laying queens. They further argue that the down-regulation of Notch signaling could be one of the first steps occurring in the ovaries of workers when a colony loses its queen. Their starting hypothesis was that Notch signaling, which is well established as a crucial factor in regulating cell-cell interactions in the germline stem-cell niche of the fruit fly could play a role as a fecundity modulator in the honey bee. As a first step they treated caged worker bees with the Notch signaling inhibitor DAPT and found that the percentage of workers with activated ovaries was increased, even in the presence of QMP. Next, they analyzed the expression of Notch target genes by RT-qPCR and in situ hybridization and they found that one of these, a bHLH2 homolog, predicts with high probability the activation status of the ovary, and hence the fertility of a queenless worker. As this could not be explained by the expression of the

canonical Notch ligands Delta and Serrate, the authors investigated the localization of the intracellular Notch domain (NICD), which actually transmits the signal to the nucleus. They used a *Drosophila* antibody against the NICD, which they validated for use in immunofluorescence detection on honey bees, and they performed in situ hybridization assays to detect the expression of *numb*, a gene encoding a protein involved in the intracellular recycling of Notch protein. By these approaches they convincingly showed that Numb is repressed in the presence of the queen, thus activating Notch signaling in the germlarium and stalling ovary activation.

Clearly, this is strong evidence for the role and deployment of a conserved signaling pathway to generate a molecular switch in the regulation of the fertility status of the female castes in the honey bee, the model organism for understanding how insect sociality functions at the molecular level. Thus, this work really presents a major advance and will direct future research on this critical issue to Darwin's dilemma. Furthermore, it brings with it a strong evo-devo perspective by illustrating how an ancient signaling pathway can be used (rewired?) to generate plasticity in social life histories.

The manuscript is well written - I thoroughly enjoyed reading it -, the experiments and statistical analyses follow gold standards (including the extensive validation of reference genes for the RT-qPCR assays, where they teach us a lesson), and the results are clearly presented in the main text and the Extended Resources section. This said, I have a few comments on sections where I think the authors should be more precise to make the experimental design clearer and where the discussion could be put in a broader context.

Major comments:

1) In the final sentence of the Summary the authors claim that they provide the first functional evidence linking ovary activity in adult worker bees to the presence of the queen. Ok, they convinced me that the Notch signaling is a crucial module for switching the fertility status of workers and allowing continuous oogenesis in queens, but it is not the first functional evidence. The first functional evidence for this was to show that removal of the queen and brood have this effect, and subsequently, that QMP could mimic the presence of the queen. Correct to say would be that they provide the first functional evidence for a conserved molecular signaling module linking ovary activity of adult worker bees with the presence of the queen.

2) I found the description of the experimental design to lack clarity on the source of the bees used in the different analyses, both in the main text and in the supplementary material. With respect to the DAPT treatment it is only clear from the supplementary material that the experiments were done with caged worker bees; this should also briefly be stated in the text, in line 79. What was completely unclear to me was the source from which the bees were obtained for the RT-qPCR, in situ and immunofluorescence analyses. Where these also caged bees (QMP-exposed), or where they from queenless and queenright hives, respectively? This makes a big difference because in queenright hives the presence of brood (brood pheromone) also has a suppressive effect on worker ovary activation (see comment below).

The description for setup of the DAPT experiments given in section 1.2 of the supplementary material is ok, but this information is lacking for all the other analyses. Just saying - section 1.1 in Extended Resources - that bees were kept following standard techniques is trivial,

there are hundreds, and similarly, just saying that hives were made queenless and workers were collected 2-4 weeks later is not sufficient. For instance, were brood frames removed to prevent emergency queen rearing and stimulate ovary activation in the workers?

Furthermore, where did the queenright workers come from, and the queens, what was their age, where they active egg layers heading colonies?

While this can be briefly commented on in the main text, to keep it short as required, all this information must be presented in the Extended Resources section so that the experiments can be fully appreciated.

I became fully aware of the importance of this issue when reading the conclusions and legend for the model in Figure 4, because in the text the inhibition of Notch signaling is attributed to QMP (not the presence of the queen or brood), whereas in the legend it is stated that numb expression is induced when the queen is removed from the hive. These are subtle differences, but they deserve being clarified.

3) I understand that in the main text the authors needed to concentrate on Notch signaling in the upper germarium to bring across the message that this is the key to understanding how workers can activate their ovaries when the repressor (queen) is removed.

Nonetheless, I found the results, which they present as supplementary material on Notch signaling in the lower germarium and in the developing follicles, equally exciting. I think these results should be given more value in the main text, also to avoid spreading the wrong idea that "Notch blocks ovary development in honey bee workers". It does so in the upper germarium, but in the lower germarium it appears to be very important for structuring the interaction between germline and follicle epithelial cells.

Without going into detail on these aspects in the main text, this could be discussed in the Extended Resources section. I am pretty sure the authors, being experts in insect embryogenesis, could give us some information here, especially on the peculiar neuralized expression pattern.

4) The focus of this study was to clarify the molecular circuitry by which queen presence/queen mandibular gland pheromone (QMP) suppress ovarian activity in worker bees. While this is clearly an important route for reducing conflict over male production in social Hymenoptera, the presence of larval brood that needs to be tended by nurse bees is equally important, and the authors cite two seminal studies in this respect. More recently, there has been a study that directly compared the effects of QMP and brood pheromone, which also should be mentioned: Traynor et al., 2014. *Behav. Ecol. Sociobiol.* 66:2059-2073.

5) Another aspect is the evolutionary history of queen pheromones. A recent study by Van Oystaeyen et al. (2014. *Conserved class of queen pheromones stops social insect workers from reproducing. Science* 343:287-290) provides insights into this aspect. This certainly could be important for interpreting the role of Notch signaling in the ovary. The idea is to bridge the still existing gap between the external signal (queen/QMP) and the ovary. Figure 4 represents this as a single blocking step, but the authors are, of course, aware that this can involve a central pathway (antennal perception/CNS-endocrine axis), or a more direct passage of QMP to and into the abdomen, as hypothesized by Naumann (1991, *Apidologie* 22:523-531). None of these pathways has been fully worked out so far, but the

identification and elucidation of ovarian Notch signaling shown here is definitely the first crucial step towards this goal.

6) Finally, it should be mentioned briefly for the non-expert reader that the focus of this study is to investigate the molecular switch mechanisms that underlie the option for (facultatively sterile) workers to become reproductive in the absence of an egg-laying (dominant) queen. The question is not on caste fate, which is determined during postembryonic development. Nonetheless, there could be a mechanistic link between these two decision processes, as both involve programmed cell death, which, on the one hand, leads to the degeneration of the majority of ovariole primordia during metamorphosis, and on the other is seen in specific regions of the germarium in adult queenright workers (see additional references below).

Minor issues

- line 89, correct to say is ovum (ova is the plural of this Latin word)
- in line 102 it is stated that the caged bees not exposed to QMP were actively laying eggs. Where, did they have an empty piece of frame where they could lay? Furthermore, correct to say here would be that they had fully developed eggs in their ovaries, because this was the parameter that was actually scored, not egg laying.
- line 164 - correct wording in sentence: found throughout
- line 219: correct to say is that eusociality requires one female caste (not one female) to have evolved reproductive dominance.... What is important here is caste fate and function, not the number of queens, as there can be more than one queen in a social insect colony (ants, wasps), even though in honey bees it is only one.
- I suggest to cite two recent papers reporting programmed cell death in adult honey bee workers exposed or not to QMP: Ronai et al., 2015, *J. Insect Physiology* 81:36-41; Ronai et al., 2016, *Molecular Biology and Evolution* 33:134-142.
- Figure 1: The symbols for the gray scales are too small

Reviewer #3 (Remarks to the Author):

The main goal of Duncan et al. is to investigate 'Reproductive Constraint' in the Honey Bee *Apis Mellifera*. Reproductive constraint in honey bee workers is thought to be conditional, such that in the presence of queen and brood, ovary activity in the workers is suppressed and do not lay viable male eggs, whereas in the absence of the queen and brood they do. The novelty of this paper is providing evidence for involvement of the Notch pathway in regulating this conditional activity of worker ovaries. This is important because it advances our knowledge of developmental basis of social harmony and cohesion in social insects. Overall, the experiments and the claims are convincing and recommend this paper for publication. Before publication, I would like the authors to address the following comments/critiques:

Detailed comments:

(1) Across the social insects, it would be helpful if the authors place the conditional reproductive constraints known in honeybees in the context of other existing constraints in social insects. Conditional reproductive constraints are one type of many constraints. For example, Sameshima and Ito (XXX; Evolution & Development) just published a paper documenting the reduction of spermathecae across ants.

(2) The authors must cite Traynor et al. (2014; Behav. Ecol. Sociobiol) paper has to be cited. It details the affect of QMP on ovarian activity (repression and activation).

(3) On line 21-23: Authors should be more specific and say that this is the first gene function evidence linking ovarian activity in adult worker bees to the presence of the queen.

(4) Figure 2: The authors must do a better job at explaining the inconsistency between qPCR data and In Situ data. They are consistent for panels A & B, meaning that the gene expression differences revealed by qPCR are consistent with the expression differences revealed by In situ hybridization. However, the differences are not consistent for panels C & D. Nowhere in the main text do the authors discuss this inconsistency, and it is important to do so.

(5) Figure 3: has the same problem of inconsistency as Figure 2. If the authors are using whole ovaries for the qPCR, then of course there will be differences. I recommend they move qPCR results to the supplemental methods. And explain clearly in the main text the differences.

(6) Authors should define the complex, and what the roles of the genes are within the complex. Are those genes the whole complex? What roles do the first 2 play versus the last 2.

(7) Extended Data Figure 6: In addition to the germarium, authors provide interesting data for expression data for the four complex genes in the later oocyte. Yet, this is never discussed or mentioned in the text. It would be interesting for the authors to provide one sentence as to what oocyte expression of these complex genes may mean.

(8) Figure 4: It would be great to provide evidence that numb is directly downstream of QMP and mediates notch signaling. It is a fairly straightforward experiment where all the pieces of the puzzles are there: they would have to treat with QMP and stain for numb queen-less workers with numb. They should see more staining in the queen-less worker. This would solidify this connection. It is not necessary for this paper to be published, but I encourage the authors to consider this seriously because for such a straightforward experiment, they could solidify the summary figure.

(9) In general text is poorly written with very long sentences through the manuscript (e.g. Lines 32-36)! They should streamline text throughout.

Authors' response to reviewers' comments:

Reviewer #1 (Remarks to the Author):

Before considering its publication, the authors need to address the following comments to further strengthen this paper.

1. First, it is important to know which cells express the Notch ligand Delta using fluorescent mRNA in situ and DNA staining. Germ cells have bigger nuclei than somatic cells.

We appreciate this suggestion by the reviewer and have previously attempted fluorescent *in situ* hybridisation for a number of genes in the honeybee ovary with varying degrees of success, we find that the colorimetric system we use in this study gives us the most reliable and reproducible results. The germ cells do have larger nuclei (e.g. Fig. 3B) than somatic cells. However, in this region of the honeybee ovary (the anterior germarium), the majority of the cells are germline derived (they are the dividing germ cells, termed cystocytes, that will ultimately give rise to the oocyte and nurse cells)¹. Large numbers of somatic cells are not observed until the mid-posterior of the germarium in honeybees¹. We have been interested for some time in finding molecular markers for the presumptive oocyte and also the germ-stem cell niche in honeybees and have yet to find a reliable molecular marker.

That we observe uniform staining for *Am-Delta* in all of the cells of the germarium for queen-less and queen-right worker ovaries allows us to conclude that the differences in Notch signalling we observe in this region of the queen-less worker ovary can not be attributed to spatial differences in *Delta* expression.

The pattern that we observe in queens is a little different, with expression throughout all the cells of the germarium, but apparent accumulation in a specific cell type. Based on the positioning of these cells relative to the terminal filament, our knowledge of the spatial organisation of cells in the ovary and the size of the nuclei (compared to the smaller size expected for somatic cells) we have determined that these are likely to be the cystocytes (dividing germ-line cells). This finding and expression pattern is consistent with previously published data for *Am-Delta* expression in queen ovaries².

We have clarified this in the text by adding the following information (Lines 253-255)

“In the queen ovary Delta RNA is expressed by all cells in the germarium but appears enriched in a specific cell type, likely the dividing germ cells (cystocyte clusters), consistent with previously published data⁴¹.

And have amended the sentence that follows to make it clear that when comparing the expression pattern between queen-less and queen-right workers and see no differences, making our rationale clearer that *Delta* expression can not account for the differences in notch activity observed between queen-right and queen-less worker ovaries. (Lines 255-258)

“Delta RNA is, however, expressed uniformly in all cells of the germarium of queen-right worker, queen-less worker ovarioles (Figure 3A) implying Delta expression does not account for the differences in Notch activity observed between queen-right and queen-less worker ovaries (Fig. 2).”

2. Second, based on the results in Figure 4, *numb* mRNA should be expressed in germ cells. The authors should use fluorescent mRNA in situ and DNA staining to verify it. In queen-less worker ovaries we observe a loss of Notch immunofluorescence throughout the anterior of the germarium (Figure 4A, middle panel), this area of the ovary is predominantly made up of dividing germ cells (cystocyte clusters) and the germ stem cell niche, with proliferating somatic cells seen toward the posterior of the germarium¹. We observe relatively uniform staining throughout this region, including the germ cells, in queen-less worker and queen ovaries. We have added an additional sentence to the text to clarify this (Lines 320-321)
“In queen-less worker bees *numb* RNA is detected in all the cells of the germarium including the dividing germ cells (cystocyte clusters).”

3. Third, the authors should clarify if the eggs laid by the DAPT-treated workers can develop normally.

The focus of this study was on the role of Notch signalling in the early processes in oogenesis rather than on oocyte maturation and possible roles in embryogenesis³ and so we did not directly measure egg laying or hatching. We did routinely find eggs in the comb during this study, but worker policing of eggs was also observed. Additionally, the experimental conditions used in this study were not conducive to egg hatching (which generally requires high humidity) and would have necessitated a lot more manipulation and disturbance of the bees. We have added this information to the main text (Lines 404-406)

“In experiments without QMP eggs were routinely observed in the comb after several days, but policing behaviour was also occasionally seen and hatched larvae were never observed in either the DAPT or control treatments.”

Reviewer #2 (Remarks to the Author):

Major comments:

1) In the final sentence of the Summary the authors claim that they provide the first functional evidence linking ovary activity in adult worker bees to the presence of the queen. Ok, they convinced me that the Notch signaling is a crucial module for switching the fertility status of workers and allowing continuous oogenesis in queens, but it is not the first functional evidence. The first functional evidence for this was to show that removal of the queen and brood have this effect, and subsequently, that QMP could mimic the presence of the queen. Correct to say would be that they provide the first functional evidence for a conserved molecular signaling module linking ovary activity of adult worker bees with the presence of the queen.

We have amended this sentence as follows (new text underlined) (Lines 21-22)

“provide evidence for the first molecular mechanism directly linking ovary activity of adult worker bees with the presence of the queen”

2) I found the description of the experimental design to lack clarity on the source of the bees used in the different analyses, both in the main text and in the supplementary material.

a. With respect to the DAPT treatment it is only clear from the supplementary material that the experiments were done with caged worker bees; this should also briefly be stated in the text, in line 79.

We now include this information in the main text (Line 121) as follows (new text underlined),

“To determine whether Notch signalling is involved in regulating ovary activity in worker honeybees, we caged newly emerged bees and treated them with DAPT (or solvent control) for 10 days in the absence of QMP.”

b. What was completely unclear to me was the source from which the bees were obtained for the RT-qPCR, in situ and immunofluorescence analyses. Where these also caged bees (QMP-exposed), or where they from queenless and queenright hives, respectively? This makes a big difference because in queenright hives the presence of brood (brood pheromone) also has a suppressive effect on worker ovary activation (see comment below).

Caged bees were only used for the DAPT experiments (Fig. 1) the remainder of the bees were derived from queen-less and queen-right hives. We have now included this to the materials and methods section (Lines 364-370) as follows (new text underlined).

“Queen-right worker bees were obtained from standard hive with a laying queen (queen-right). To obtain queen-less worker bees frames with brood and bees were removed from queen-right hives and placed into a standard nucleus box. Colonies regularly surveyed for queen cells (which were destroyed) and the presence of worker laid eggs. Ovaries were dissected from queen-less worker bees 2 - 4 weeks after establishing a queen-less hive; a sufficient period for brood to have emerged and worker laid eggs were detected. Stages of oogenesis were based on an established scheme².”

c. The description for setup of the DAPT experiments given in section 1.2 of the supplementary material is ok, but this information is lacking for all the other analyses. Just saying - section 1.1 in Extended Resources - that bees were kept following standard techniques is trivial, there are hundreds, and similarly, just saying that hives were made queenless and workers were collected 2-4 weeks later is not sufficient. For instance, were brood frames removed to prevent emergency queen rearing and stimulate ovary activation in the workers? Furthermore, where did the queenright workers come from, and the queens, what was their age, where they active

egg layers heading colonies? While this can be briefly commented on in the main text, to keep it short as required, all this information must be presented in the Extended Resources section so that the experiments can be fully appreciated.

We have now included this information to the materials and methods section (Lines 358-370) as follows (new text underlined).

“A. mellifera were cultured using standard techniques in Dunedin, New Zealand¹. Honeybees were housed in wooden Langstroth hives or wooden nucleus boxes at three sites around Dunedin. All bees were originally obtained from Betta Bees Research Limited. Queen tissue was obtained from young queens (~12 months old) that were active egg layers heading colonies. All Queen-right worker bees were obtained from standard hive with a laying queen (queen-right). To obtain queen-less worker bees, frames with brood and bees were removed from queen-right hives and placed into a standard nucleus box. Colonies were regularly surveyed for queen cells (which were destroyed) and the presence of worker laid eggs. Ovaries were dissected from queen-less worker bees 2 - 4 weeks after establishing a queen-less hive; a sufficient period for brood to have emerged and worker laid eggs were detected.”

We have also added the following information to the main text (Lines 165-168)

“For the experiments that follow, queen tissue was obtained from young queens (~12 months old) that were active egg layers heading colonies, queen-right worker bees were obtained from standard hive with a laying queen and queen-less worker bees were obtained from a manipulated hive that contained a queen and no young brood.”

d. I became fully aware of the importance of this issue when reading the conclusions and legend for the model in Figure 4, because in the text the inhibition of Notch signaling is attributed to QMP (not the presence of the queen or brood), whereas in the legend it is stated that numb expression is induced when the queen is removed from the hive. These are subtle differences, but they deserve being clarified.

We have amended the text to refer to “the queen” rather than QMP, as the experiments that point to Numb having a possible role in this process were not done in cages with synthetic QMP, they were done in natural hives and so we have corrected this error in the text to link this to the presence of the queen rather than QMP directly. We have added information to the main text (refer to point c above) about the use of caged bees versus bees from a natural hive, and have included more detail in the Materials and Methods (point b above) as to how we obtained queen-less worker bees, which we believe further clarifies this point.

3) I understand that in the main text the authors needed to concentrate on Notch signaling in the upper germarium to bring across the message that this is the key to understanding how workers can activate their ovaries when the repressor (queen) is removed. Nonetheless, I found the results, which they present as supplementary material on Notch signaling in the lower germarium and in the developing follicles, equally exciting. I think these results should be given more value in the main text, also to avoid spreading the wrong idea that "Notch blocks ovary development in honey bee workers". It does so in the upper germarium, but in the lower germarium it appears to be very important for structuring the interaction between germline and follicle epithelial cells.

We have included the following sentence in the main text (new text underlined) and refer the reader to the appropriate Supplementary Data (Lines 293-296)

“(protein (as seen in follicle cells late in oogenesis, Supplementary Fig. S4, where immunoreactivity for the Notch protein and expression of genes of the E(spl)-C may indicate that Notch signalling has a role in supporting active oogenesis, specifically by patterning the follicle cells (refer to Supplementary section 2.3))”

Without going into detail on these aspects in the main text, this could be discussed in the Extended Resources section. I am pretty sure the authors, being experts in insect embryogenesis, could give us some information here, especially on the peculiar neuralized expression pattern.

We have added some additional information to the Supplementary data section (Lines 359-362) as follows (new text underlined).

“Mid-way through oogenesis *neuralized* RNA becomes enriched at the dorsal surface of the oocyte and this enrichment persists as the oocyte matures (Supplementary Fig. 5F). This expression pattern is similar to that seen for *tailless* in honeybees which has a role in patterning early honeybee embryos⁴². During oogenesis *tailless* RNA is localised to the dorsal surface enriching in the posterior as the oocyte matures⁴². During early embryogenesis maternal *tailless* RNA moves to the posterior of the oocyte⁴² and RNA interference studies have shown that it has a role in posterior patterning⁴². The similar localization of *neuralized* RNA to the dorsal surface of the developing oocyte may indicate that it has novel functions in axis formation or embryonic patterning. A previous study, however, demonstrated that Notch signalling does not have a role in segmentation in the honeybee¹⁰”.

4) The focus of this study was to clarify the molecular circuitry by which queen presence/queen mandibular gland pheromone (QMP) suppress ovarian activity in worker bees. While this is clearly an important route for reducing conflict over male production in social Hymenoptera, the presence of larval brood that needs to be tended by nurse bees is equally important, and the authors cite two seminal studies in this respect. More recently, there has been a study that directly compared the effects of QMP and brood pheromone, which also should be mentioned: Traynor et al., 2014. Behav. Ecol. Sociobiol. 66:2059-2073.

We have now included a citation to this study (Line 55).

5) Another aspect is the evolutionary history of queen pheromones. A recent study by Van Oystaeyen et al. (2014. Conserved class of queen pheromones stops social insect workers from reproducing. Science 343:287-290) provides insights into this aspect. This certainly could be important for interpreting the role of Notch signaling in the ovary. The idea is to bridge the still existing gap between the external signal (queen/QMP) and the ovary. Figure 4 represents this as a single blocking step, but the authors are, of course, aware that this can involve a central pathway (antennal perception/CNS-endocrine axis), or a more direct passage of QMP to and into the abdomen, as hypothesized by Naumann (1991, Apidologie 22:523-531). None of these pathways has been fully worked out so far, but the identification and elucidation of ovarian Notch signaling shown here is definitely the first crucial step towards this goal.

We appreciate the reviewers comment that our study is a first crucial step in linking the external signal (QMP / presence of the queen) with ovary activity. We agree that there is likely an intermediate signalling step between the brain/antennae and the ovary and think that having a fixed

output of the pathway (e.g. the effect on Notch signalling on the ovary) will help us to determine how QMP is controlling reproduction in the worker honeybee. We have included a new sentence (Lines 344-347) to make this clear to the reader

“It is yet to be determined whether QMP is directly affecting the ovary or acting via signalling between the brain or antennae”

The Van Oystaeyen paper referred to by the reviewer is interesting as it implies that there is a conserved class of queen pheromones that stop social insect workers from reproducing. It is worth noting though that, compared with the other social hymenopterans surveyed, the honeybee queen pheromone is quite derived (with much higher levels of higher levels of fatty acids, unsaturated hydrocarbons and esters compared with other species). Without knowing exactly which component of QMP is affecting Notch signalling in the honeybee ovary it is difficult to speculate whether this may be a conserved mechanism of controlling reproduction via queen pheromones in the hymenoptera.

6) Finally, it should be mentioned briefly for the non-expert reader that the focus of this study is to investigate the molecular switch mechanisms that underlie the option for (facultatively sterile) workers to become reproductive in the absence of an egg-laying (dominant) queen. The question is not on caste fate, which is determined during postembryonic development. Nonetheless, there could be a mechanistic link between these two decision processes, as both involve programmed cell death, which, on the one hand, leads to the degeneration of the majority of ovariole primordia during metamorphosis, and on the other is seen in specific regions of the germarium in adult queenright workers (see additional references below).

We have tried to make the focus of this study clearer by including new text as follows (underlined) (Lines 60-66)

“The phenotypic differences between queen and worker ovaries are established during larval development in response to larvae being fed royal jelly. This nutritional stimulus initiates distinct developmental trajectories in larvae resulting in the morphologically distinct queen and worker castes and the reduced reproductive capacity of workers. Worker bees have a reduced reproductive capacity, they have no spermatheca (sperm storage organ) and their ovaries are smaller¹⁸.”

We have also included references to the papers on programmed cell death mentioned by the reviewer, but as we did not address programmed cell death in our paper we have not included more detail or discussion of these papers.

Minor issues –

line 89, correct to say is ovum (ova is the plural of this Latin word)

As we are talking about “ovaries” (plural) we believe our use of ova is correct.

in line 102 it is stated that the caged bees not exposed to QMP were actively laying eggs.

Where, did they have an empty piece of frame where they could lay? This information is now included in the Materials and Methods section.

Furthermore, correct to say here would be that they had fully developed eggs in their ovaries, because this was the parameter that was actually scored, not egg laying.

This has been amended.

line 164 - correct wording in sentence: found throughout

This has now been amended in response to Reviewer 1. The sentence now reads “***Delta RNA is, however, expressed uniformly in all cells of the germarium of queen-right worker, queen-less worker ovarioles (Figure 3A)***”

line 219: correct to say is that eusociality requires one female caste (not one female) to have evolved reproductive dominance.... What is important here is caste fate and function, not the number of queens, as there can be more than one queen in a social insect colony (ants, wasps), even though in honey bees it is only one.

This has been amended to read “one female caste”

I suggest to cite two recent papers reporting programmed cell death in adult honey bee workers exposed or not to QMP: Ronai et al., 2015, J. Insect Physiology 81:36-41; Ronai et al., 2016, Molecular Biology and Evolution 33:134-142.

We have included citations for these papers.

Figure 1: The symbols for the gray scales are too small

We have made these larger.

Reviewer #3 (Remarks to the Author):

(1) Across the social insects, it would be helpful if the authors place the conditional reproductive constraints known in honeybees in the context of other existing constraints in social insects. Conditional reproductive constraints are one type of many constraints. For example, Sameshima and Ito (XXX; Evolution & Development) just published a paper documenting the reduction of spermathecae across ants.

We have clarified that we are referring specifically to honeybees by specifying in our introduction (L39)

“In honeybees reproductive constraints are both behavioural or physiological; examples...”

It would be really interesting to do an in depth analysis of reproductive constraints in eusocial hymenopterans, but we feel that such an analysis is beyond the scope of this manuscript. I assume the manuscript the reviewer is referring to is Gotoh et al., 2016 (as I couldn't find a manuscript by Sameshima and Ito addressing this topic). Gotoh et al., has done an extensive analysis on the development of spermatheca in ant species and we now include a citation to this study in our introduction (Line 29)

(2) The authors must cite Traynor et al. (2014; Behav. Ecol. Sociobiol) paper has to be cited. It details the affect of QMP on ovarian activity (repression and activation). We have now cited this paper in our introduction (Line 55).

(3) On line 21-23: Authors should be more specific and say that this is the first gene function evidence linking ovarian activity in adult worker bees to the presence of the queen. (Line 21) We have now amended this to read “and provide the first molecular mechanism directly linking ovary activity in adult worker bees with the presence of the queen”

(4) Figure 2: The authors must do a better job at explaining the inconsistency between qPCR data and In Situ data. They are consistent for panels A & B, meaning that the gene expression differences revealed by qPCR are consistent with the expression differences revealed by In situ hybridization. However, the differences are not consistent for panels C & D. Nowhere in the main text do the authors discuss this inconsistency, and it is important to do so.

In the main text we highlight the main reason that we think there is a discrepancy (Lines 196-212), as follows

“As our quantitative RT-PCR experiments (Fig. 2) were performed on RNA derived from whole ovaries and are thus an amalgamation

of RNA expression in all ovarian cell types, it was important to identify the region/s in which Notch signalling is acting in the ovaries to constrain reproduction (full expression patterns are shown in Supplementary Fig. 2). To determine which cells in the honeybee ovary are receiving the Notch signal we used the expression of the E(spl)-C genes as a proxy for Notch activity, using *in situ* hybridization to visualize cells expressing RNAs from these genes (Fig. 2, Supplementary Fig. 2).

We now include a further sentence directly addressing this perceived discrepancy (Lines 220-225), new text underlined.

“All four genes of the E(spl)-C are also expressed in the posterior of the ovariole as the oocyte matures, with RNA detected in the nurse and follicle cells (Supplementary Fig. 2). The expression of these genes in regions of the ovary other than the germarium may explain why, using RT-qPCR on whole ovaries, the expression of bHLH2 and Her decreases dramatically as worker ovaries initiate oogenesis while the other two genes of the complex do not respond (Fig. 2).”

(5) Figure 3: has the same problem of inconsistency as Figure 2. If the authors are using whole ovaries for the qPCR, then of course there will be differences. I recommend they move qPCR results to the supplemental methods. And explain clearly in the main text the differences.

We feel strongly that the two techniques we have used to assess gene expression in the honeybee ovary are complementary, and both provide useful relevant information. RT-qPCR provides quantitative gene expression data and gives the reader an appreciation for the magnitude of changes in gene expression that we observe in these ovaries. We then, as stated in the text, use *in situ* hybridization (which is not quantitative), to determine which cells-and cell-types these genes are expressed in. We have now included extra information to the readers (see response to 4, above) which will make it clear why we don't expect to see 100% agreement between our RT-qPCR and *in situ* hybridisation data and that these techniques are presenting different but allied data.

(6) Authors should define the complex, and what the roles of the genes are within the complex. Are those genes the whole complex? What roles do the first 2 play versus the last 2.

We have added the requested information as follows (Lines 212-216).

“The E(spl)-C is complex of Notch responsive genes that is evolutionarily conserved amongst insects and crustaceans^{43,44}. In most insects and crustaceans the complex consists of three basic helix-loop-helix transcription factors and a bearded class protein^{43,44}. In honeybees it is unclear what the functions of these four genes are⁴³.”

We also include a possible explanation for the differential responsiveness of these genes in honeybee based on what is known about the regulation of E(spl)-C genes in *Drosophila* (Lines 182-186).

“Studies of the *Drosophila* E(spl)-C indicate that genes within the complex are regulated by discrete enhancers and can be differentially responsive to Notch signalling⁴⁴⁻⁴⁶. Similar differences in regulation may also account for the different expression of bHLH2 and Her, but not bearded or bHLH1 in the honeybee ovary (Figure 2).”

(7) Extended Data Figure 6: In addition to the germarium, authors provide interesting data for expression data for the four complex genes in the later oocyte. Yet, this is never discussed or mentioned in the text. It would be interesting for the authors to provide one sentence as to what oocyte expression of these complex genes may mean.

In response to reviewer 2 we have included the following sentence in the main text (please note that we have amended the order of figures in the supplementary material to reflect the order mentioned in the main text. The figure referred to by the reviewer (Extended data Figure 6) is now Supplementary Figure 4.

“(as seen in follicle cells late in oogenesis, Supplementary Fig. S4, where immunoreactivity for the Notch protein and expression of genes of the E(spl)-C may indicate that Notch signalling has a role in supporting active oogenesis, specifically by patterning the follicle cells (refer to Supplementary section 2.3))

We had previously mentioned the possible role of maternal expression in Extended Data section 2.3 (Lines 239-241)

“Maternal provision of these RNAs to the developing oocyte may suggest a possible role for these genes in regulating oocyte maturation or early developmental processes.”

(8) Figure 4: It would be great to provide evidence that *numb* is directly downstream of QMP and mediates notch signaling. It is a fairly straightforward experiment where all the pieces of the puzzles are there: they would have to treat with QMP and stain for *numb* queen-less workers with *numb*. They should see more staining in the queen-less worker. This would solidify this connection. It is not necessary for this paper to be published, but I encourage the authors to consider this seriously because for such a straightforward experiment, they could solidify the summary figure.

This is analogous to the experimental data that we have presented in the manuscript, where we used *in situ* hybridization to show that *Numb* RNA is only detectable in the germarium of worker bees in the absence of the queen.

The reviewer is suggesting using queen-less worker bees (which have active oogenesis and high levels of *numb* expression) and treating with QMP to determine if *numb* expression decreases. Although this would be an interesting experiment I am not convinced this experiment would give an unequivocal answer, as there are still many things unknown about the way the ovary responds to QMP. For instance, we do not know if queen-less bees that are then exposed to QMP ‘regress’ and stop egg laying, the speed at which this might happen, or if it is dependent on the physiological state of the ovary (for instance whether ovaries at stage 1 (prior to yolk deposition) might regress faster than those at stage 3). To do this experiment with queen-less workers and get an interpretable result you would ideally need to know what stage of ovary development the bees are when starting the experiment, which is not possible as phenotyping is a destructive process.

Also, if *numb* expression doesn’t decrease this doesn’t necessarily mean that *numb* is not downstream of QMP as it is possible that once Notch signalling is turned off in the germarium a positive feedback loop is established that maintains ovary activity.

We ideally need to experimentally modulate *numb* expression using RNA interference or another technology in the honeybee ovary. Alternatively, we need to understand the molecular regulation of the *numb* gene in honeybee to show that *numb* is directly downstream of QMP. We think this is very interesting experiment! It is unfortunately not currently possible to do the experiment in such a way as to get an unequivocal result and may not add anything to the manuscript. What we have shown is that *numb* transcription is responsive to the loss of the queen, and that *numb* RNA is expressed in queen-right worker ovaries in the region of the ovary where we see degradation of the Notch protein (based on immunofluorescence).

We think that it is likely that QMP is affecting *numb* expression through multiple mechanisms and that this need not be the result of a direct effect of QMP on *numb* transcription. We have clarified this by adding the following sentence to the manuscript (Lines 344-347)

“It is yet to be determined whether QMP is directly affecting the ovary or acting via signalling between the brain or antennae.”

We have also modified our model (Fig. 4D) such that a dashed-line links QMP with *numb* expression, to indicate this point.

(9) In general text is poorly written with very long sentences through the manuscript (e.g. Lines 32-36)! They should streamline text throughout.

We have taken the opportunity to review the manuscript extensively and have edited the manuscript paying particular attention to shortening long sentences.

References

- 1 Tanaka, E. D., Schmidt Capella, I. C. & Hartfelder, K. Cell death in the germline - mechanisms and consequences for reproductive plasticity in social bees. *Brazilian journal of morphological sciences* **23** (2006).
- 2 Wilson, M. J., Abbott, H. & Dearden, P. K. The evolution of oocyte patterning in insects: multiple cell-signaling pathways are active during honeybee oogenesis and are likely to play a role in axis patterning. *Evolution & development* **13**, 127-137, doi:10.1111/j.1525-142X.2011.00463.x (2011).
- 3 Wilson, M. J., McKelvey, B. H., van der Heide, S. & Dearden, P. K. Notch signaling does not regulate segmentation in the honeybee, *Apis mellifera*. *Development genes and evolution* **220**, 179-190, doi:10.1007/s00427-010-0340-6 (2010).

REVIEWERS' COMMENTS:

Reviewer #1 (Remarks to the Author):

The authors have addressed the comments from this reviewer. Thus, this exciting story is ready for publication.

Reviewer #2 (Remarks to the Author):

All my comments made on the Original version of this manuscript were satisfactorily addressed and clarified in the current revised version and/or the Responses to the Reviewers letter. I have no further comments.

Reviewer #3 (Remarks to the Author):

The authors have made great progress in addressing reviewer concerns and have greatly improved the manuscript from its original version. The manuscript undeniably has the potential to have great impact on our understanding of reproductive division of labor and social evolution. Here are some minor comments for the authors to consider for further improvement of the manuscript prior to publication:

1. The authors should speculate as to why Notch signaling in *Apis mellifera* has opposite functions to Notch signaling in other invertebrates and vertebrates (Xu and Gridley, 2012). For example, in *Drosophila* oogenesis, Notch, Delta and Serrate positively regulate oogenesis and are required for wild type function of ovarioles. If an average reader were looking at how the authors cite previous literature on the function of Notch in oogenesis they would have the false impression that it reinforces the results shown by the authors in this manuscript, when in fact they are contradicting. It is known that Delta and Serrate expressed on the surface of germline stem cells (GSC) activate Notch in somatic cells to form and maintain the GSC niche, which induces and maintains stem cell fate. This is crucial for the asymmetric cell division in the anterior of the germarium for oogenesis to proceed and allow continuous egg production. Additionally, loss of Delta in the germline results in fused egg chambers and inhibition of Notch signaling through DAPT treatment in mice increases oocyte apoptosis (Feng et al. 2014) Speculating that Notch, which already plays a key role in oogenesis in insects and mammals, has been coopted to instead inhibit oogenesis in an adapted process to maintain a functionally sterile worker caste would increase the impact of these results. It is not uncommon for cell signaling pathways to have context dependent functions where they can both promote and inhibit the same process under different conditions or cellular contexts and the idea that perhaps the Notch pathway has been selected upon to generate a non-reproductive caste by inhibiting oogenesis rather than its canonical roles in promoting oogenesis is exciting but the authors much directly state this possibility. Down regulation of Notch signaling and lack of NCID in the early germarium of queen and queen-less workers honeybees is completely novel and exciting, however, without discussing how this deviates with everything that is known about Notch

signaling in the broader context of oogenesis literature the potential impact of this finding is lost. The authors state in their conclusion that Notch signaling has been coopted from solitary insects to restrain reproduction in Apis but this has no impact if they do not indicate from the onset the nature of Notch signaling in other organisms, which is positive rather than negative regulation. The authors attempt to rationalize that Notch signaling has gone from nutritionally responsive in flies to socially in honeybees and while this is interesting, the differences are much more dramatic than that, as outlined above Notch signaling is a fundamental component of oogenesis. Without addressing this, it is not even clear what the basis was at the onset to study Notch as a negative regulatory of oogenesis in Apis, as it is a positive regulator in other invertebrates and vertebrates.

2. In line 123 the authors state that DAPT treatment can partially overcome QMP activity on workers yet from line 126-129 the authors state that this shows that inhibition of Notch signaling can overcome QMP activity and is the key mechanism by which QMP acts to inhibit oogenesis. This is an overstatement, it should read "partially" overcomes QMP activity on line 127 and "may be a key mechanism" by which QMP regulates oogenesis. An increase in egg laying activity from 5% to 15% does not indicate that 1) inhibition of Notch signaling overcomes QMP effects and 2) That Notch is the key mechanism by which QMP acts, especially given that the baseline of egg laying without QMP treatment is 28% (line 120)

3. Line 66-67As the authors describe Notch signaling as pleiotropic, ancient and conserved, it would be of use to general readers to understand what other processes notch signaling is known to regulate eg. Embryonic development, cell fate specification, and stem cell maintenance. Too broad to just describe it as a cell-signaling pathway. It is noted that they describe this in the beginning of the results section but this may be more appropriate in the introduction.

4. Line 74 should read "Notch signaling is 'a' proximate mechanism" There can be other upstream factors at play, especially as authors admit that it is unclear if QMP acts on ovaries directly or through signaling from brain and/or antennae

5. Result subheadings would be more appropriate as statements rather than questions.

Authors' response to reviewers' comments:

Reviewer #1 (Remarks to the Author):

The authors have addressed the comments from this reviewer. Thus, this exciting story is ready for publication.

Reviewer #2 (Remarks to the Author):

All my comments made on the Original version of this manuscript were satisfactorily addressed and clarified in the current revised version and/or the Responses to the Reviewers letter. I have no further comments.

Reviewer #3 (Remarks to the Author):

The authors have made great progress in addressing reviewer concerns and have greatly improved the manuscript from its original version. The manuscript undeniably has the potential to have great impact on our understanding of reproductive division of labor and social evolution. Here are some minor comments for the authors to consider for further improvement of the manuscript prior to publication:

1. The authors should speculate as to why Notch signaling in *Apis mellifera* has opposite functions to Notch signaling in other invertebrates and vertebrates (Xu and Gridley, 2012). For example, in *Drosophila* oogenesis, Notch, Delta and Serrate positively regulate oogenesis and are required for wild type function of ovarioles. If an average reader were looking at how the authors cite previous literature on the function of Notch in oogenesis they would have the false impression that it reinforces the results shown by the authors in this manuscript, when in fact they are contradicting. It is known that Delta and Serrate expressed on the surface of germline stem cells (GSC) activate Notch in somatic cells to form and maintain the GSC niche, which induces and maintains stem cell fate. This is crucial for the asymmetric cell division in the anterior of the germarium for oogenesis to proceed and allow continuous egg production. Additionally, loss of Delta in the germline results in fused egg chambers and inhibition of Notch signaling through DAPT treatment in mice increases oocyte apoptosis (Feng et al. 2014) Speculating that Notch, which already plays a key role in oogenesis in insects and mammals, has been coopted to instead inhibit oogenesis in an adapted process to maintain a functionally sterile worker caste would increase the impact of these results. It is not uncommon for cell signaling pathways to have context dependent functions where they can both promote and inhibit the same process under different conditions or cellular contexts and the idea that perhaps the Notch pathway has been selected upon to generate a non-reproductive caste by inhibiting oogenesis rather than its canonical roles in promoting oogenesis is exciting but the authors much directly state this possibility. Down regulation of Notch signaling and lack of NCID in the early germarium of queen and queen-less workers honeybees is completely novel and exciting, however, without discussing how this deviates with everything that is known about Notch signaling in the broader context of oogenesis literature the potential impact of this finding is lost. The authors state in their conclusion that Notch signaling has been coopted

from solitary insects to restrain reproduction in Apis but this has no impact if they do not indicate from the onset the nature of Notch signaling in other organisms, which is positive rather than negative regulation. The authors attempt to rationalize that Notch signaling has gone from nutritionally responsive in flies to socially in honeybees and while this is interesting, the differences are much more dramatic than that, as outlined above Notch signaling is a fundamental component of oogenesis. Without addressing this, it is not even clear what the basis was at the onset to study Notch as a negative regulatory of oogenesis in Apis, as it is a positive regulator in other invertebrates and vertebrates.

The reviewer raises an excellent point, and we had previously decided not to discuss the details of Notch regulation in *Drosophila* in order to keep the manuscript relatively short. However, we agree that discussion of this would strengthen the manuscript and have revised the discussion to include this point (Lines 340-415). New text is included in bold below.

“Repression of oogenesis via Notch signalling in the germaria of worker bee ovaries is consistent with the low levels of Notch activity we observe in the queen ovaries (Fig. 2-4) where oogenesis is actively occurring¹⁸. The association of active Notch signalling with repression of oogenesis contrasts with roles for Notch signalling in controlling reproduction in *Drosophila*³⁰. In *Drosophila*, Notch signalling both establishes and maintains the germ stem cell niche^{25,31}. Increased Notch signalling results in a larger niche and more stem cells^{25,55} but it is unclear if this role is ancestral and representative of all insects. Notch signalling has other roles in *Drosophila* oogenesis including promoting the differentiation of follicle cells³⁰ but in other insects^{56,57}, Notch signalling inhibits follicle cell differentiation, implying that the ancestral function of Notch signalling in insect follicle cells is to maintain undifferentiated cell fates^{56,57}. Our data may indicate that the ancestral function of Notch signalling in the germarium was also to repress cell differentiation.

Our results show that Notch signalling has a pivotal role in repressing reproduction in the worker honeybee in the presence of the queen and her pheromone. By enforcing the reproductive division of labour and mitigating conflict over male production, Notch signalling is critical to the reproductive constraints that underpin the evolution of eusociality. Notch signalling, presumably as a result of adaptive evolution, has been co-opted from a role in solitary insects into constraining reproduction in the worker honeybee. Notch signalling’s fundamental role in the ovary, known to be environmentally responsive⁵⁸, has been modified and transformed in honeybees into social control of reproduction in the adult honeybee.”

2. In line 123 the authors state that DAPT treatment can partially overcome QMP activity on workers yet from line 126-129 the authors state that this shows that inhibition of Notch signaling can overcome QMP activity and is the key mechanism by which QMP acts to inhibit oogenesis. This is an overstatement, it should read "partially" overcomes QMP activity on line 127 and "may be a key mechanism" by which QMP regulates oogenesis. An increase in egg laying activity from 5% to 15% does not indicate that 1) inhibition of Notch signaling overcomes QMP effects and 2) That Notch is the key mechanism by which QMP acts, especially given that the baseline of egg laying without QMP treatment is 28% (line 120)

We have amended L126 (new line 194) to add the word “This **finding** indicates that **partial** inhibition of Notch...” and have changed “a” to “**may be a**” on L128 (new Line 180).

3. Line 66-67As the authors describe Notch signaling as pleiotropic, ancient and conserved, it would be of use to general readers to understand what other processes notch signaling is known to regulate eg. Embryonic development, cell fate specification, and stem cell maintenance. Too broad to just describe it as a cell-signaling pathway. It is noted that they describe this in the beginning of the results section but this may be more appropriate in the introduction.

We now include the following paragraph in the introduction (Lines 89-100)

“Notch signalling is pivotal during embryogenesis and in adult animals to control processes such as differentiation and cell fate specification and, depending on the biological context, proliferation and apoptosis^{23,24}. Notch signalling is typified by its role in specification of neuronal versus epidermal cells during neurogenesis in *Drosophila*, but Notch signalling has a role in the development of most tissues and organs in many animals²³. In *Drosophila*, Notch signalling has multiple roles in oogenesis and reproduction; for instance Notch signalling is responsible for specifying the germ cell niche²⁵, controls proliferation and differentiation of somatic follicle cells²⁶ and defines distinct follicle cell populations²⁷”

4. Line 74 should read "Notch signaling is 'a' proximate mechanism" There can be other upstream factors at play, especially as authors admit that it is unclear if QMP acts on ovaries directly or through signaling from brain and/or antennae
We have taken the reviewers suggestion and altered L74 to read “**a proximate mechanism**”

5. Result subheadings would be more appropriate as statements rather than questions. Result sub-headings have been changed in accordance with this reviewer and at the request of editorial staff.